# Multi-omic signatures of host response associated with presence, type, and outcome of enterococcal bacteremia

Charlie Bayne,[1,2,3] Dominic McGrosso,[1,2,3] Concepcion Sanchez,[1,2,3] Leigh-Ana Rossitto,[1,2,3] Maxwell Patterson,[2] Carlos Gonzalez,[2,3] Courtney Baus,[4] Cecilia Volk,[5] Haoqi Nina Zhao,[3] Pieter Dorrestein,[2,3,6,7] Victor Nizet,[2,3,6] George Sakoulas,[6,8] David J. Gonzalez,[2,3,7] Warren Rose[5]

**ABSTRACT** Despite the prevalence and severity of enterococcal bacteremia (EcB), the mechanisms underlying systemic host responses to the disease remain unclear. Here, we present an extensive study that profiles molecular differences in plasma from EcB patients using an unbiased multi-omics approach. We performed shotgun proteomics and metabolomics on 105 plasma samples, including those from EcB patients and healthy volunteers. Comparison between healthy volunteer and EcB-infected patient samples revealed significant disparities in proteins and metabolites involved in the acute phase response, inflammatory processes, and cholestasis. Several features distinguish these two groups with remarkable accuracy. Cross-referencing EcB signatures with those of *Staphylococcus aureus* bacteremia revealed shared reductions in cholesterol metabolism proteins and differing responses in platelet alpha granule and neutrophil-associated proteins. Characterization of *Enterococcus* isolates derived from patients facilitated a nuanced comparison between EcB caused by *Enterococcus faecalis* and *Enterococcus faecium,* uncovering reduced immunoglobulin abundances in *E. faecium* cases and features capable of distinguishing the underlying microbe. Leveraging extensive patient metadata, we now have identified features associated with mortality or survival, revealing significant multi-omic differences and pinpointing histidine-rich glycoprotein and fetuin-B as features capable of distinguishing survival status with excellent accuracy. Altogether, this study aims to culminate in the creation of objective risk stratification algorithms—a pivotal step toward enhancing patient management and care. To facilitate the exploration of this rich data source, we provide a user-friendly interface at https://gonzalezlab.shinyapps.io/EcB_multiomics/.

**IMPORTANCE** *Enterococcus* infections have emerged as the second most common nosocomial infection, with enterococcal bacteremia (EcB) contributing to thousands of patient deaths annually. To address a lack of detailed understanding regarding the specific systemic response to EcB, we conducted a comprehensive multi-omic evaluation of the systemic host response observed in patient plasma. Our findings reveal significant features in the metabolome and proteome associated with the presence of infection, species differences, and survival outcome. We identified features capable of discriminating EcB infection from healthy states and survival from mortality with excellent accuracy, suggesting potential practical clinical utility. However, our study also established that systemic features to distinguish *Enterococcus faecalis* from *Enterococcus faecium* EcB show only a moderate degree of discriminatory accuracy, unlikely to significantly improve upon current diagnostic methods. Comparisons of differences in the plasma proteome relative to healthy samples between bacteremia caused by *Enterococcus* and *Staphylococcus aureus* suggest the presence of bacteria-specific responses alongside conserved inflammatory reactions.

**Peer Reviewer** Małgorzata Barbara Łobocka, Institute of Biochemistry and Biophysics, Polish Academy of Sciences, Warsaw, Poland

Address correspondence to Warren Rose, warren.rose@wisc.edu, or David J. Gonzalez, djgonzalez@health.ucsd.edu.

P.D. is a scientific advisor and holds equity in Cybele and bileOmix, and he is a scientific co-founder and advisor and holds equity in Ometa, Arome, and Enveda with prior approval by UC-San Diego.

See the funding table on p. 23.

KEYWORDS proteomics, metabolomics, *Enterococcus*, multi-omics, bacteremia, microbiology

Enterococci are widely distributed in the environment and have co-evolved as common microorganisms within the gastrointestinal microbiota since terrestrial animals first transitioned from water to land (1). From this diverse lineage, the distantly related species *Enterococcus faecalis* and *Enterococcus faecium* have independently evolved to become members of the human gut microbiome. These microbes establish their niche as commensals within the first 10 days following birth (2, 3) and typically compose <0.1% of the gut microbiome (4). Under certain circumstances, *E. faecalis* and *E. faecium* can become pathogenic. These conditions are commonly encountered in healthcare settings, where several features acquired during their evolution have allowed *E. faecalis* and *E. faecium* to become important healthcare-associated pathogens (5).

These microbes possess the inherent ability to survive commonly used disinfection routines and persist on surfaces in healthcare settings, thereby facilitating patient-to-patient transmission (6). Additionally, they are increasingly resistant to antibiotics, owing both to an intrinsic resistance to commonly used broad-spectrum antibiotics, such as cephalosporins and carbapenems (7), and to an impressive capacity to acquire mobile genetic elements through horizontal gene transfer, which enhances their fitness (8). Even more concerning is that newly developed antibiotics targeting Gram-positive pathogens are either ineffective against enterococci or rapidly lead to the emergence of resistance (9–12). Furthermore, antibiotics that are active against enterococci, such as beta-lactams in the case of *E. faecalis*, are bacteriostatic, necessitating prolonged courses of combination antibiotic therapy to prevent relapse (13). These traits, combined with inadequate antibiotic stewardship and the increase in aggressive medical treatments in an aging patient population, have led to a rise in severe, invasive infections with a mortality rate of 25%–50% in enterococcal bacteremia (EcB) (6, 7, 9–11, 14).

Host factors are widely recognized as critical determinants of the outcome of host-microbe interactions (15) and have been used as prognostic biomarkers to predict patient outcomes in a variety of diseases, ranging from coronavirus disease 2019 (COVID-19) to cancer (16, 17). While molecular features associated with mortality have been described in *Staphylococcus aureus* bacteremia (18), in the context of EcB, the prediction of mortality has so far been limited to crude clinical metrics such as severity of illness and age (19). A comprehensive profile of the molecular features of the systemic response in a well-documented EcB patient cohort would enable the discovery of associations with successful and suboptimal outcomes.

To describe the systemic host response in EcB patients, we employed high-resolution tandem mass tag (TMT) LCMS3 mass spectrometry (MS)-based proteomics and metabolomics to profile plasma samples collected from clinical EcB cases and healthy controls. Furthermore, we utilized previously published results from our group to compare deviations from homeostasis observed in EcB and *S. aureus* bacteremia. This data set provides an initial assessment of the ability to use unbiased molecular features of the host response to predict the presence of EcB, whether the bacteremia is caused by *E. faecalis* or *E. faecium*, and the outcome of the infection. The enhanced understanding provided by this multi-omic resource can serve as a starting point for developing novel therapeutic strategies aimed at improving patient outcomes in EcB.

## MATERIALS AND METHODS

### Experimental design and statistical rationale

The study conducted here utilized human plasma collected from patients between 2018 and 2021 at UW Health, a 450-bed tertiary academic medical center in Madison, WI. The sample size was determined based on technical considerations for multiplexed proteomics and metabolomics approaches, as well as logistical constraints (20, 21).

Further details regarding statistical analysis, demographics, and clinical data are provided in the sections below.

## Human plasma samples

Upon admission, plasma was obtained from 32 patients with *E. faecium* bacteremia and 44 patients with *E. faecalis* bacteremia. Enterococcal bacteremia was diagnosed through positive blood cultures and treated with antibiotics. *E. faecalis* and *E. faecium* bacteremia were differentiated usingmatrix assisted laser desorption ionization-time of flight (MALDI-TOF)-based rapid identification and confirmed using standard culture and biochemical methods. Additionally, plasma from 29 healthy controls was collected from blood bank volunteers.

## Clinical data collection

Patient electronic medical records were reviewed to collect basic demographics, including age, gender, and comorbidities. Data on the infection and treatment (antibiotic use and source control methods) and clinical course included the organism type (*E. faecalis*/*E. faecium*), source of the bloodstream infection (endovascular, urine, abdominal fluid, etc.), antibiotic susceptibility, and laboratory values and markers of infection (e.g., serum creatinine, white blood cell count with differential, temperature, vital signs). The mean age of the patients was 59.6 ± 16.3 years, and 59% were male. Among the 76 patients with enterococcal bacteremia, 30.1% were infected with vancomycin-resistant *Enterococcus*, as confirmed by routine antimicrobial susceptibility testing in the clinical microbiology laboratory. The total duration of bacteremia included cases of persistent bacteremia (consecutive days of positive blood cultures) and in-hospital microbiologic relapse, defined as the recurrence of a positive blood culture after the first negative culture while receiving appropriate antibiotic therapy. The mean duration of bacteremia was 2.7 ± 1.9 days, with a median of 2 days and a range of 1–14 days. The mortality rate during hospitalization and within 1 year of infection onset was 21.7% and 38.6%, respectively.

## Proteomics arm

### Protein preparation

A 25 µL aliquot of plasma from each patient was added to 200 µL of lysis buffer containing 6 M urea, 7% SDS, 50 mM tetraethylammonium bromide (TEAB), one protease inhibitor tablet (Roche cat # 06538282001), and one PhosStop tablet (Roche cat # 04906845001), with the final pH adjusted to 8.1 using phosphoric acid. The samples were then sonicated in a water bath and vortexed for 5 min each. Disulfide bonds were reduced using 100 µM dithiothreitol (DTT) for 30 min at 47°C, followed by brief cooling on ice. Alkylation was performed with 300 µM iodoacetamide for 45 min at room temperature in the dark. The reactions were quenched with 100 µM DTT at room temperature for 5 min. Samples were loaded onto S-Trap 96-well plates (Protifi) per the manufacturer's instructions. Samples were then digested with 5 µg of trypsin in 115 µL 50 mM TEAB for 3 hours at 47°C. Peptides were eluted sequentially with 125 µL of 50 mM TEAB, then 5% formic acid (FA), and finally 50% acetonitrile with 5% FA and were then dried under vacuum centrifugation. The peptides were desalted using 50 mg Sep-Pak tC-18 cartridges (Waters cat # WAT054960) per manufacturer instructions and were quantified using a Pierce Quantitative Colorimetric Peptide Assay (ThermoFisher cat # 23275). A 50 µg aliquot of each sample was designated for TMT labeling, with an additional 10 µg from each sample combined into a final 50 µg aliquot for use as the bridge channel.

## TMT labeling

Aliquots were labeled with TMT-Pro 16-Plex reagents (cat # A44520, ThermoFisher Scientific, lot # WJ327115) as described previously with channel 134N reserved as the bridge channel (22).

## Chromatography and mass spectrometry

Basic pH reverse-phase LC, followed by data acquisition through LC–MS2/MS3, was performed as previously described (22). Briefly, 75-min linear gradients of 22% to 35% acetonitrile and 10 mM ammonium bicarbonate were used to elute peptides on HPLC C18 columns (Biobasic). The resulting 96 fractions were combined as previously described (23). These fractions were then analyzed using tandem mass spectrometry (MS2/MS3) on an Orbitrap Fusion mass spectrometer (ThermoFisher) equipped with an in-line EASY-nLC 1000 (ThermoFisher). Separation and acquisition settings were performed using previously defined methods (24).

## Statistics and data analysis

Thermo .raw files were converted to mzML files using MSConvert, with a peak picking (centroiding) filter applied to the conversion. The resulting mzML files were then input into Fragpipe version 20.0 and analysis was performed using MSFragger version 3.8, IonQuant version 1.9.8, and Philosopher version 5.0.0. Fragpipe was run using the TMT-16 MS3 quantification (TMT16-MS3) workflow. Briefly, this workflow specifies the use of DDA mode with low mass accuracy MS2 (Ion trap) for identification and quantification utilizing a high mass accuracy (Orbitrap) MS3. MSFragger settings included a precursor mass tolerance of 20 ppm for both upper and lower limits, strict trypsin digestion rules allowing for two missed cleavages, and the search was set to include static modifications of carbamidomethylation of cysteines, TMTpro on lysines and N termini, and variable modifications of oxidized methionine. A database containing the human reference proteome (UP000005640) concatenated with the reference proteomes from *Enterococcus faecalis* (UP000001415) and *Enterococcus faecium* (UP000321556) was used to match spectral IDs. All reference proteomes were downloaded from UniProt on 11 September 2023.

Proteins identified as differentially abundant were subjected to gene ontology (GO) analysis using all proteins identified in the experiment as the universe, as implemented in ClusterProfiler4.0 (25). All GO terms described have adjusted $P$-values ≤0.05 unless otherwise described.

## Metabolomics arm

### Plasma metabolite extraction

All steps were done on ice unless otherwise indicated. Plasma samples (50 µL) were thawed for 30 min, and then 200 µL of prechilled extraction solvent (100% methanol with 1 mM sulfamethazine as an internal standard) was added to each sample. Samples were mixed by vortexing for 2 min and then incubated at 20°C for 20 min to aid in protein precipitation. Samples were centrifuged at 16,000 × $g$ for 15 min to pellet the protein precipitate. The supernatant was then transferred into a 96-well deep well plate, dried using a centrifugal low-pressure system, and stored at −80°C once dry.

### Metabolomics LC-MS2 analysis

Metabolomics LC-MS2 was performed on a Q-Exactive mass spectrometer coupled to a Thermo Vanquish HPLC system. The chromatographic analysis was carried out on a Polar C18 100A LC Column 100 × 21 mm (catalog no. 00D-4759-AN). 5 µL of each sample was injected and run on a 10-min gradient. The mobile phase solvents (solvent A, water – 0.1% formic acid; solvent B, acetonitrile – 0.1% formic acid) were run at a flow rate of 0.500 mL/min, and chromatographic separation was achieved using the following

gradient: 0 to 1 min 5% B; 1 to 7 min a linear increase from 5 to 100% B; 7 to 7.5 min held at 100% B; 7.5 min to 8 min a linear decrease from 100% to 5% B; and then 5% B from 8 min to 10 min.

## Metabolite identification

Full scan MS spectra (*m/z* 100–1,500) were acquired and the top 5 most intense ions from a unique scan were fragmented. Dynamic exclusion was set to 10.0 s. The isolation window was set to 3.0 *m/z* with an isolation offset of 0.5 *m/z*, and the intensity threshold was set to 5e4.

## Raw file processing

Thermo .raw files were first converted to mzML using MSConvert (3.0.22155-0ff594) (26). mzML files were processed together using MzMine (3.5.0) (27) to identify metabolite features. Parameters for individual sample metabolite feature identification were as follows: mass detection (MS1 noise level: 1.0E3, MS2 noise level: 5.0E2), feature detection through ADAP chromatogram builder (min group size in # of scans = 4, group intensity threshold = 3,000, min highest intensity = 1,000, *m/z* tolerance = 0.005 Da or 10 ppm), feature detection chromatogram resolving (MS/MS scan pairing with retention time (RT) tolerance = 0.10 min and MS1–MS2 precursor tolerance = 0.0100 *m/z*; local min search used with chromatographic threshold = 90%, min RT range 0.50 min, min relative height 0.01%, min absolute height 1,000, min ratio of peak top/edge = 1.7, peak duration 0.05–1 min, and min # data points = 4), 13C isotope filter (*m/z* tolerance = 0.01 *m/z*, RT tolerance = 0.30 min, and maximum charge = 5). Parameters for the metabolome feature bucket table were as follows: join aligner (*m/z* tolerance = 0.01 *m/z*, *m/z* weight = 80, RT tolerance = 0.30 min, RT weight = 20), feature list filtering (at least two peaks per row), and gap filling (intensity tolerance = 10%, *m/z* tolerance = 0 *m/z*, RT tolerance = 0.4). Metabolite feature tables were then exported using the GNPS FBMN option to generate the required files for online GNPS FBMN analysis.

## Feature annotations

Feature annotation done through the Global Natural Products Social Molecular Networking (GNPS) feature-based molecular networking (FBMN) workflow (version 28.2) (28). MS2 MGF files were exported from MzMine3 along with a feature quantification table and imported to the FBMN workflow. Spectral library files were set to search all GNPS spectral libraries. Library search min matched peaks were set to 6 and the score threshold to 0.7. Precursor mass tolerance was set to 0.02 Da and fragment ion mass tolerance to 0.02 Da. Features with annotation were then appended to the feature quantitation table exported from MzMine.

## Molecular networking

A molecular network was created using the online workflow (https://ccms-ucsd.github.io/GNPSDocumentation/) on the GNPS website (http://gnps.ucsd.edu). The data were filtered by removing all MS/MS fragment ions within ±17 Da of the precursor *m/z*. MS/MS spectra were window filtered by choosing only the top 6 fragment ions in the ±50 Da window throughout the spectrum. The precursor ion mass tolerance was set to 2.0 Da and an MS/MS fragment ion tolerance of 0.5 Da. A network was then created where edges were filtered to have a cosine score above 0.7 and more than six matched peaks. Furthermore, edges between two nodes were kept in the network if and only if each of the nodes appeared in each other's respective top 10 most similar nodes. Finally, the maximum size of a molecular family was set to 100, and the lowest scoring edges were removed from molecular families until the molecular family size was below this threshold. The spectra in the network were then searched against GNPS' spectral libraries. The library spectra were filtered in the same manner as the input data. All

matches kept between network spectra and library spectra were required to have a score above 0.7 and at least six matched peaks.

### Data processing of metabolite features

Metabolomics data were normalized using sulfamethazine as a single internal standard as described previously (29). Briefly, any feature containing a 0 value in any sample was discarded, and values for each remaining feature were divided by the observed value for sulfamethazine within each sample. This resulting value was then log10 transformed and multiplied by 1E6.

### Biomarker identification

An ensemble feature selection approach, which combines the Mann-Whitney U-test, Pearson and Spearman correlations, logistic regression, and four variable importance measures derived from two different implementations of the random forest algorithms *cforest* and *randomforest,* was employed as an R package to select features with minimal bias (30).

### Metadata assessment

Metadata correlations were assessed in the following manner. First, categorical metadata associations were determined using Mann-Whitney *U* or Kruskall-Wallis tests when appropriate. Continuous metadata associations were determined using Pearson correlation. Associations in the figures represent the $-\log10$ (*P*-value) of each test. All tests were performed in R.

### Statistical analysis

All statistical analyses were completed as reported in the corresponding figure legends or methods details. R was used to conduct all tests. For all tests, significance is denoted as follows: ****$P < 0.0001$; ***$P < 0.001$; **$P < 0.01$; *$P < 0.05$, ns—not significant.

### Machine learning analysis

Machine learning analysis was performed using the tidymodels framework in R (31). Models built with proteomics and metabolomics data were restricted to putatively annotated metabolites and features that had no missing values across the 105 samples included in the study. Models built using the clinical metadata excluded race, death_at_one_year, peripheral_vascular_disease, dementia hemiplegia, highly_active_antiretroviral_therapy, and source, as these columns had limited variability across our set of patients.

Lasso regression models were trained on 80% of the data, using fivefold cross-validation to optimize and evaluate model performance. In cases where multiple models achieved identical performance on the training set, the first reported model was selected. These models were then evaluated on the remaining 20% of the data, which served as an unseen test set. Performance on the test set was visualized using receiver operator characteristic (ROC) curves, and estimates of the model coefficients were provided.

## Nanopore sequencing

### Strain isolation

*E. faecium* and *E. faecalis* clinical isolates were recovered from blood culture vials by plating patient blood onto solid media. Single isolated colonies were then recovered and inoculated into liquid culture and once cultures reached turbidity, a final concentration of 15% glycerol was added, and stocks were frozen at −80°C. Strain identification was differentiated via MALDI-TOF rapid identification and was confirmed using traditional culture-based and biochemical methods.

## DNA extraction and QC

*Enterococcus faecium* and *Enterococcus faecalis* from the frozen glycerol stocks were inoculated into 1 mL of brain heart infusion (BHI) broth. These cultures were subsequently incubated at 37℃ shaking at 220 RPM overnight. Bacteria were pelleted by centrifugation at 5,000 × *g* for 5 min, after which supernatant was discarded and the pellets were resuspended in a modified lysis buffer containing 1 mL of QIAGEN B1 buffer supplemented with 2.29 mg/mL lysozyme, 0.29 mg/mL Labiase (cat no: OZ-30EX OZEKI Ci., Ltd.), and 0.2 mg/mL RNAaseA (cat # 1007885 QIAGEN). Labiase was added due to our observation that lysozyme alone was insufficient to lyse many of these clinical strains, due to well-documented lysozyme resistance among enterococcal clinical isolates (32). These cells were then incubated overnight at 37℃ to facilitate lysis. The next day, 45 µL of proteinase K solution (cat # RP107B-10 QIAGEN) was added to each sample and incubated for 1 hour at 37℃. Next, 0.35 mL of QIAGEN Buffer B2 was added to each sample, tubes were mixed several times by inversion and then incubated at 50℃ for 30 min. The resulting high molecular weight DNA was then purified using QIAGEN Genomic-tip 20/G (cat # 10223 Qiagen). A genomic tip was equilibrated with 1 mL of buffer QBT. Samples were vortexed for 10 s at maximum speed and then applied to the equilibrated genomic tips. After all liquid had passed through, each genomic tip was washed three times with 1 mL of QIAGEN Buffer QC. Genomic DNA was then eluted by applying 1 mL of Buffer QF twice. Also, 1.4 mL of room temperature isopropanol was then added to the eluate, and it was inverted several times to precipitate the DNA. To collect the DNA, samples were centrifuged at 12,000 × *g* for 15 min at 4℃. The supernatant was then carefully removed, and then the DNA pellet was washed with 1 mL of 70% ethanol. The samples were then vortexed briefly and then centrifuged at 12,000 × *g* for 10 min at 4℃. This wash and centrifugation step was then repeated. The supernatant was then removed, taking great care not to disturb the pellet, and was air-dried for 10 min before being resuspended in 50 µL of 10 mM Tris-Cl, pH 8.5. The DNA was then dissolved by shaking at room temperature overnight, followed by gentle pipetting with a wide bore pipette tip.

The resulting DNA was then checked for purity by Nanodrop and DNA concentration was assessed using Qubit dsDNA Quantification Assay Kit, broad range (ThermoFisher cat # Q32853). A subset of purified DNA samples was run on a Genomic DNA ScreenTape (Agilent, cat # 5365) to assess DNA length and integrity.

## Barcoding and pooling

Four hundred nanograms of purified DNA from each enterococcal strain was barcoded using Native Barcoding Kit 96 V14 (Oxford Nanopore cat # SQK-NBD114.96). The manufacturer's ligation sequencing genomic DNA (gDNA) – Native Barcoding Kit 96 V14 version NBE_9171_v114_revl_15Sep2022 protocol was followed as described to perform DNA repair and end prep, native barcode ligation, adapter ligation and clean up, priming and loading the SpotOn flow cell.

## Data acquisition

A 10.4.1 flow cell was loaded with ~20 fmol pooled barcoded DNA (estimated from the library concentration combined with the TapeStation-derived distribution of molecular weight). Minknow 23.07.5 was used to acquire the data in high accuracy basecalling mode, facilitated by a custom-built desktop computer equipped with a Nvidia GeForce RTX 3070 GPU.

## Sequencing analysis

Unwanted *Streptomyces fulvissimus* reads present in our Labiase preparation were bioinformatically decontaminated by mapping all reads to a reference database using centrifuge (33)and reads mapping to streptomyces were then removed using seqtk (34). Decontaminated reads were then used as input into the Epi2me labs graphic interface

which was used to run the wf-bacterial-genomes nextflow workflow (35). Briefly, this workflow concatenates input files and generates per read stats via bamstats (36), performs *de novo* assembly via flye (37), polishes with medaka (38), performs multi locus sequence typing (39) to infer the identity of the isolate, and lastly performs antimicrobial resistance calling via ResFinder (40) to identify genes associated with antimicrobial resistance. Basic assembly statistics such as N50 and the number of contigs were also generated using assembly_stats (41). The resulting FASTA assemblies were then used as input to Roary (42) to generate a pangenome.

## RESULTS

### Overview of multi-omic EcB patient plasma analysis

Using a comprehensive multi-omic strategy, we aimed to profile the host response to EcB (Fig. 1A). We analyzed plasma samples from cohorts consisting of 29 healthy volunteers, 32 patients diagnosed with *E. faecium* bacteremia, and 44 patients with *E. faecalis* bacteremia. Extensive clinical metadata were collected corresponding to each EcB patient, including gender, the Charlson comorbidity index, duration of bacteremia, day of blood draw, and mortality during admission (Fig. 1B). Additionally, antimicrobial susceptibility testing was performed on the enterococcal isolates obtained from each patient. As expected, based on the reported susceptibilities of clinical isolates, all *E. faecalis* isolates were resistant to vancomycin, while only a minority of *E. faecium* isolates displayed vancomycin resistance (43). This highlights that initial identification of the species can inform the selection of optimal antibiotic therapy for EcB.

In the TMT-guided proteomics data set, we quantified a total of 589 proteins mapping to the human proteome, with 278 proteins quantified across all samples. No proteins were found to map to the *Enterococcus* proteomes. In our metabolomics data set, we quantified a total of 10,092 features, with 693 of these features being putatively identified based on their MS2 profiles using GNPS molecular networking. Of these putatively identified metabolites, 212 were quantified across all samples. This comprehensive profiling enabled us to examine the differences between healthy individuals and those infected with *E. faecalis* or *E. faecium*, while also providing a unique opportunity to explore associations with mortality vs. survival.

Unsupervised hierarchical clustering of both the global proteomics and metabolomics data revealed a clear delineation between the healthy and infected groups (Fig. 1C and D). Significant differences between *E. faecalis* and *E. faecium* bacteremia were observed at the level of individual features (see Fig. 4A and 5A), yet the global dissimilarities between the two types of bacteremia were more subtle. This indicates large-scale similarity in the host response elicited by these two related pathogens.

### Differences in plasma profiles of bacteremia types compared to healthy

We initially focused on identifying proteins most effective at differentiating EcB from healthy populations. Compared to healthy volunteers, 204 proteins were identified to be more abundant in individuals with EcB, while 85 proteins were significantly less abundant (Fig. 2A, with false discovery rate (FDR) adjusted *P*-value ≤0.05). Some proteins showed highly significant Benjamini-Hochberg adjusted *P*-values, reaching as low as $1 \times 10^{-25}$. Similarly, evaluation of the metabolomics data revealed that 427 features were significantly increased in the infected group compared to the healthy group, while 968 features were significantly decreased. FDR adjusted *P*-values for these features were as low as $1 \times 10^{-50}$ (Fig. 3A). Of the metabolite features identified in our study, 29% of them were putatively identified using GNPS molecular networking (Fig. 3B). This partial annotation highlights a well-documented limitation inherent to current untargeted metabolomics analysis approaches (44).

To further understand the differences between these bacteremia types, we investigated how many of the specific proteins identified as significantly altered relative to healthy were shared among the different types of bacteremia. Additionally, we compared these

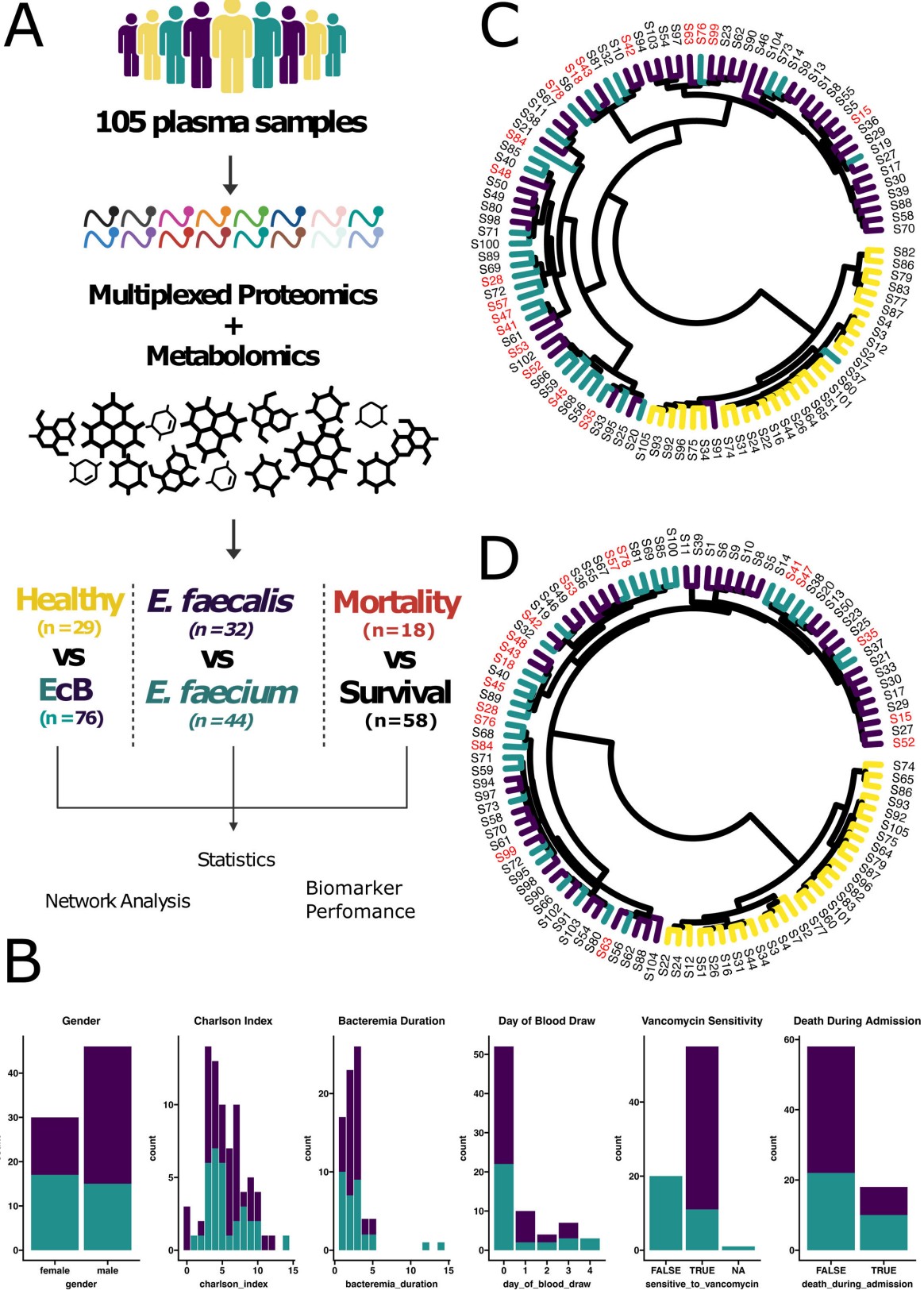

**FIG 1** Multi-omic analysis of enterococcal bacteremia patient plasma. (A) Workflow for enterococcal bacteremia plasma analysis. (B) Distribution of values collected from selected clinical metadata fields across enterococcal bacteremia patients. (C) Unsupervised hierarchical clustering of proteomics data as visualized after calculating Euclidean distance and utilizing the Ward.D2 agglomeration method. Colors of branches indicate the species of infection (yellow = healthy

Fig 1 (Continued)

volunteer, navy = *E. faecalis*, teal = *E. faecium*) and color of sample ID labels represent mortality or survival (black = survival, red = mortality). (D) Unsupervised hierarchical clustering of metabolomics data as visualized after calculating Euclidean distance and utilizing the Ward.D2 agglomeration method. Colors of branches indicate the species of infection (yellow = healthy volunteer, navy = *E. faecalis*, teal = *E. faecium*) and color of sample ID labels represent mortality or survival (black = survival, red = mortality).

deviations from homeostasis observed upon infection to another clinically relevant pathogen by analyzing previously published proteomic differences between *S. aureus* bacteremia patients and healthy volunteers (18). We found that 24% (53) of the significant proteins that increased upon infection were common across all types of bacteremia, while 13% (30) were specific to *E. faecium* bacteremia, none were specific to *E. faecalis* bacteremia, and 42% (93) were specific to *S. aureus* bacteremia (Fig. 2D). In terms of proteins that were found to be significantly decreased in infections, 28% (94) were shared across all types of bacteremia, 2% (6) were specific to EcB, 3% (10) specific to *E. faecium* bacteremia, 2% (7) to *E. faecalis* bacteremia, and 32% (106) were found only in *S. aureus* bacteremia (Fig. 2D). This analysis underscores the complex and often overlapping protein expression profiles in bacterial infections and highlights specific protein markers that could potentially differentiate between these infections.

We explored the biological processes involving the proteins identified as statistically significant in comparisons between infected and healthy individuals, focusing on how these processes varied among bacteremia caused by *Enterococcus*, *E. faecalis*, *E. faecium*, and *S. aureus*. To facilitate this analysis, we conducted GO enrichment analysis on the proteins identified as significantly different in binary comparisons (Fig. S5). For both types of EcB, we observed an enrichment in biological processes such as neutrophil chemotaxis, tertiary granule lumen, focal adhesion, extracellular exosome, and inflammatory response (Fig. 2B). These findings indicate that despite the microbial differences, there are common host responses involving critical immune and structural cellular responses.

Enterococcal and *S. aureus* bacteremia share several biological processes that are significantly depleted upon infection. Notably, proteins involved in cholesterol metabolism were reduced in infected patients, as evidenced by the significant reduction in GO terms such as reverse cholesterol transport, cholesterol homeostasis, cholesterol metabolic process, very low-density lipoprotein particle, high-density lipoprotein particle, and blood microparticle. Similarly, we observed commonalities in processes related to blood clotting between EcB and *S. aureus* bacteremia, with significant depletion noted in the GO terms for blood coagulation, heparin binding, and zymogen activation. Of note, the platelet alpha granule lumen was the only GO term observed to have the opposing effects in different types of bacteremia. It was found to be enriched in the infected samples from EcB but depleted in those from *S. aureus* bacteremia (Fig. 2B).

The metabolites that were significantly differentially abundant (adjusted *P*-value ≤0.05) underwent enrichment analysis to determine if any class of molecule was significantly enriched. Interestingly, steroids were found to be significantly enriched in infected patients compared to all features detected in the experiment (Fig. 3C). A closer examination revealed that 10 out of the 12 molecules annotated as steroids could be more specifically described as bile acids. These included the primary bile acid cholic acid, the conjugated primary bile acids glycocholic acid (GCA), taurocholic acid (TCA), glycochenodeoxycholate (GCDCA), glycochenodeoxycholic acid, glycohyocholic acid, taurochenodeoxycholic acid as well as the conjugated secondary bile acids tauroursodeoxycholic acid (TUDCA), taurohyodeoxycholic acid, and glycoursodeoxycholic acid. The same functional enrichment was observed in *S. aureus*-infected samples (Fig. S2), where the abundances of TCA, GCA, TUDCA, 3beta-hydroxy-5-cholenoic acid, and 12-ketodeoxycholic acid were enriched. In addition, several phosphatidylcholine molecules were identified as being significantly depleted in *S. aureus* bacteremia (Fig. S2). While this class of molecules did not rise to the level of statistical significance in GO enrichment analysis from healthy to EcB patients, we did note that several phosphatidylcholines were

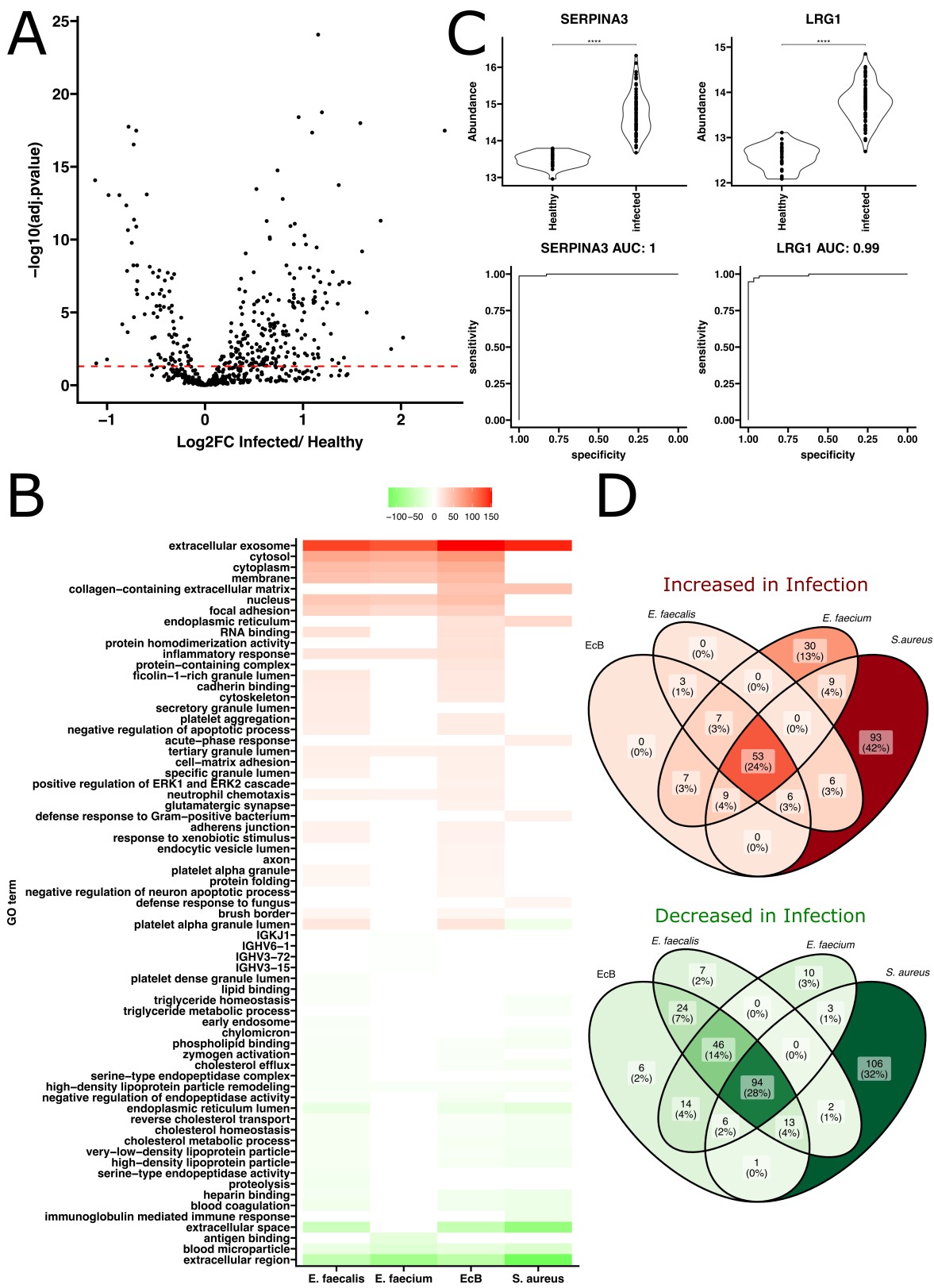

**FIG 2** Untargeted proteomics reveal dramatic differences between bacteremia types and healthy plasma. (A) Volcano plot comparing log2 fold change and FDR adjusted *P*-values of protein abundances observed when comparing infected to healthy. (B) Significantly enriched GO terms from plasma proteomics of patients suffering from enterococcal, *E. faecalis*, *E. faecium*, and *S. aureus* bacteremia. (C) Top 2 performing protein biomarkers as ranked using ensemble feature selection. (Continued on next page)

Fig 2 (Continued)

Violin plot statistics indicate results of *t*-tests. (D) Venn diagram displaying the numbers of significantly different proteins (FDR adjusted *P*-value ≤ 0.05) shared between enterococcal, *E. faecalis*, *E. faecium*, and *S. aureus* bacteremia when compared to healthy.

also significantly depleted in the case of EcB bacteremia (Table S2). This highlights a potentially common lipid metabolism disruption in bacteremia, regardless of the bacterial species involved.

We next set out to evaluate the potential utility of features within our data set as biomarkers to distinguish healthy from infected samples. To rank these biomarkers, we used ensemble feature selection (EFS), an unbiased approach that integrates outcomes from eight distinct feature selection algorithms (30, 45), subsequently aggregating and assigning ranks to the scores. This approach helps mitigate many of the inherent biases often associated with individual algorithms (45, 46). We set the correlation threshold at 0 in the EFS, ensuring that well-performing biomarkers that were highly correlated with others would still be ranked highly.

Our analysis identified the top-ranked protein and metabolite biomarkers—serpin A3-1 (SERPINA3), leucine-rich alpha-2-glycoprotein (LRG1), threonylcarbamoyladenosine, and 13-OXO-ODE—as highly effective at distinguishing infected from non-infected samples. When evaluated using logistic regression, these biomarkers produced area under the curve (AUCs) ranging from 0.99 to 1.0, indicating nearly perfect discrimination (Fig. 2C and 3D). Other top biomarkers identified in our study also were able to distinguish healthy and EcB plasma extremely well, as shown in Fig. S4, highlighting the extreme differences between the metabolic and proteomic profiles of healthy and infected individuals. A complementary machine learning model using Lasso regression showed similar performance when evaluated on an unseen test set, achieving a receiver operating characteristic (ROC) AUC of 1 (Fig. S11). There was notable overlap in the features important to the model and those identified with the alternate EFS-based approach (Fig. S5 and S11).

To benchmark these findings, we evaluated two well-known clinical biomarkers of inflammation, C-reactive protein (CRP) and serum amyloid A (SAA1), which are commonly used to monitor infection progression. Both proteins were significantly elevated in infected samples compared to healthy controls (Fig. S3). As expected, there were no significant differences in the levels of these proteins between samples infected with *E. faecalis* and *E. faecium*. When analyzed using the EFS approach, these clinically validated biomarkers demonstrated good performance, though they were not among the top performers in our data set, ranking 12th and 47th, respectively. Subsequent ROC analysis confirmed their ability to differentiate infected from uninfected samples with a high degree of sensitivity and specificity as expected, displaying AUC values of 0.97 for CRP and 0.92 for SAA1 (Fig. S3). These results reaffirm the reliability of CRP and SAA1 as indicators of infection while highlighting the potential of using EFS to discover additional biomarkers that could further enhance diagnostic accuracy.

Due to the critical role of cytokines as modulators of the immune system, we examined potential differences in cytokine profiles by bacteremia type within our data set. Given that cytokines are not readily detected in plasma using untargeted MS-based proteomics due to their low levels of absolute abundance (47), we utilized a previously reported method to infer cytokine profiles. This method leverages known interactions between proteins and cytokines to infer cytokine abundance (18). Applying this technique, we found that the inferred level of Tumor necrosis factor alpha (TNF-α) was significantly increased in EcB-infected samples compared to healthy samples, with an adjusted *P*-value ≤0.05 (Fig. S6). Additionally, we observed a trend indicating an increase in many other cytokines in infected samples compared to healthy samples, with interleukin 6 (IL-6) being notably higher, but they did not reach the statistical significance threshold set at an alpha level of 0.05.

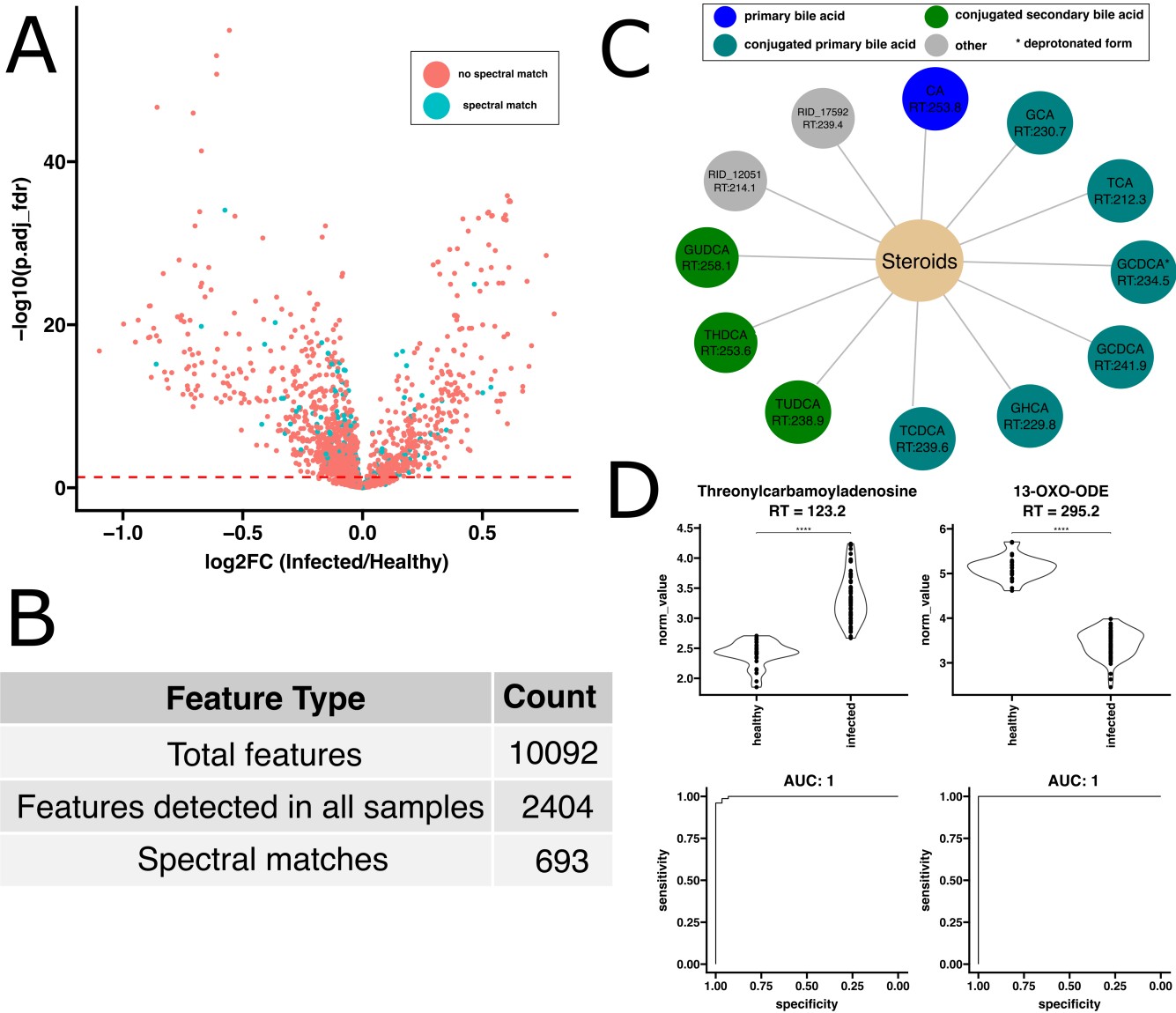

**FIG 3** Untargeted metabolomics reveal dramatic differences between enterococcal bacteremia and healthy plasma. (A) Volcano plot comparing log2 fold change and FDR adjusted *P*-values resulting from *t*-tests of normalized metabolite abundances comparing infected to healthy. No spectral match (red points) indicates features that did not produce a spectral match to any of the metabolites in the GNPS database, while spectral match (blue points) indicates features that produced a spectral match to metabolites in the GNPS database. (B) Count of the number of metabolite features identified in this study. (C) Enrichment analysis of metabolite spectral matches found to be significantly different ($P_{adj}$ ≤0.05) in infected patients relative to healthy. Colors show bile acid class (blue = primary bile aid, teal = conjugated primary bile acid, and green = conjugated secondary bile acid). RID12051 and RID17592 are features corresponding to spectral matches to (((3 a,6b,7b)-3,6,7-trihydroxy-24-oxocholan-24-yl)amino)ethanesulfonic acid and (((3 a,12b)-3,12-dihydroxy-24-oxocholan-24-yl)amino)etha-nesulfonic acid, respectively. (D) Evaluation of the top 2 performing metabolite spectral matches as ranked using ensemble feature selection for distinguishing enterococcal bacteremia patients from healthy. Violin plot statistics indicate results of *t*-tests.

## Differences in systemic plasma profiles between EcB caused by *E. faecalis* and *E. faecium*

To further delineate the systemic plasma profiles caused by different enterococcal species, we focused on cases of EcB where >90% of the isolates were identified as either *E. faecalis* or *E. faecium*. We identified 70 proteins that were significantly enriched (adjusted *P*-values < 0.05) in plasma from patients infected with *E. faecalis* compared to those with *E. faecium*. Conversely, 30 proteins were significantly enriched in *E. faecium* relative to *E. faecalis* (Fig. 4A). Similarly, in the metabolomics data, we found 11

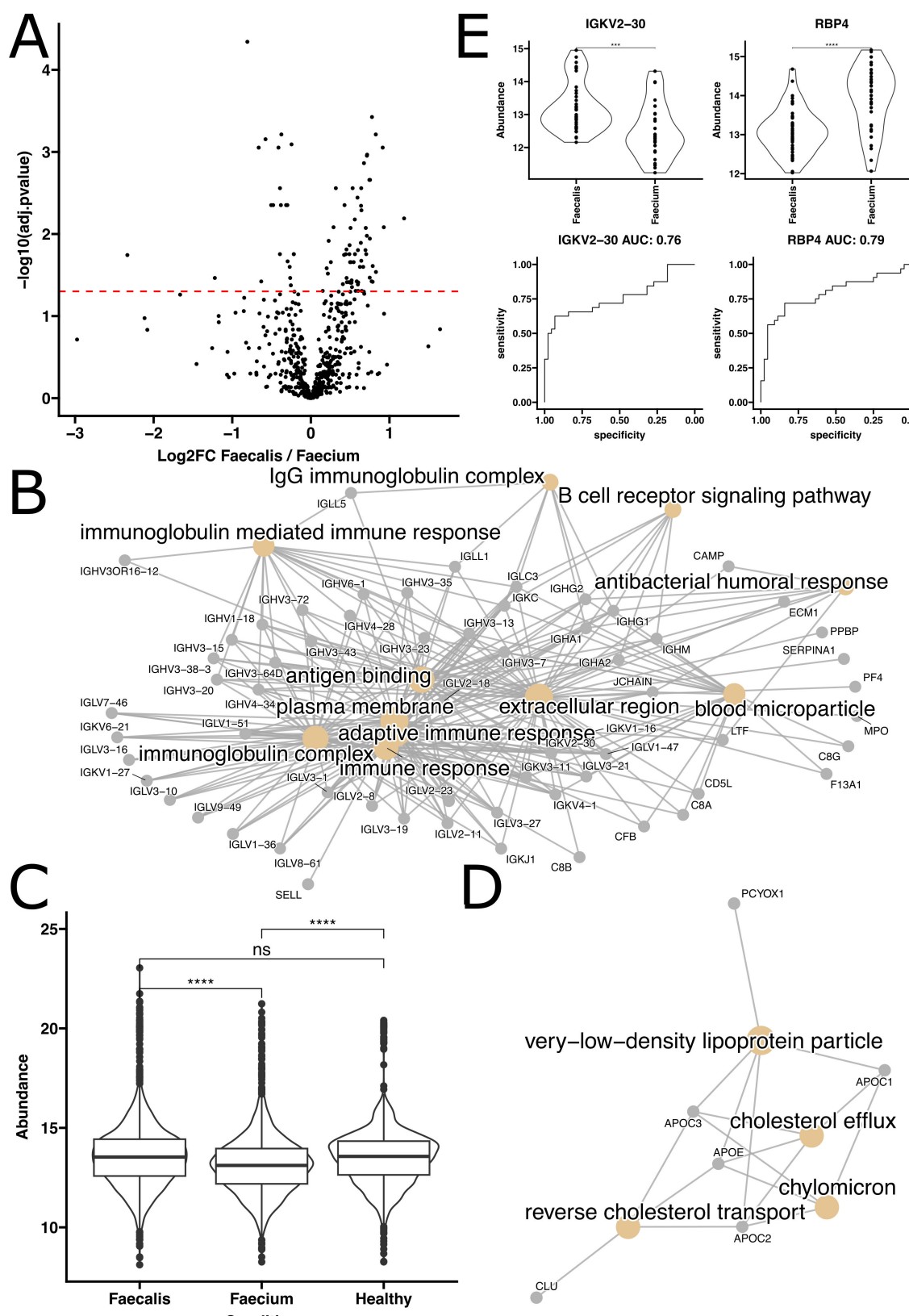

**FIG 4** Untargeted proteomics reveals differences between *Enterococcus faecalis* and *Enterococcus faecium* bacteremia. (A) Volcano plot comparing log2 fold change and FDR adjusted *P*-values of protein abundances observed when comparing *E. faecalis*- and *E. faecium*-infected patients. (B) GO Term enrichment analysis of the proteins found to be significantly enriched (FDR adjusted *P*-value ≤ 0.05) in proteins significantly more abundant in *Enterococcus faecalis* patients (Continued on next page)

Fig 4 (Continued)

relative to *Enterococcus faecium*. (C) Immunoglobulin abundances compared across patients with *E. faecalis* or *E. faecium* bacteremia and healthy volunteers. Statistics indicate result of *t*-tests adjusted for multiple comparisons. (D) GO term enrichment analysis of the proteins found to be significantly enriched in proteins significantly more abundant in *Enterococcus faecium* patients relative to *Enterococcus faecalis*. (E) Top 2 performing protein biomarkers as ranked using ensemble feature selection for distinguishing patients infected with *Enterococcus faecalis* from those infected with *Enterococcus faecium*. Violin plot statistics indicate results of *t*-tests.

metabolites significantly increased in *E. faecalis* relative to *E. faecium*, and 33 significantly increased in *E. faecium* relative to *E. faecalis* (Fig. 5A).

The GO term enrichment analysis of proteins significantly increased in *E. faecalis*-infected samples revealed dramatic differences in immunoglobulin abundances (Fig. 4B). Upon further investigation, it was found that these increased abundances *in E. faecalis* relative to *E. faecium* were attributable to reduced levels of antibodies in *E. faecium*-infected samples, as the levels of immunoglobulins in *E. faecalis* infections and healthy samples were comparable (Fig. 4C). These differential abundances of immunoglobulins were also apparent in the GO term analysis comparing *E. faecium*-infected samples to healthy ones, where there was a notable reduction in immunoglobulin-related GO terms (Fig. 2B; Fig. S2).

When analyzing the GO term enrichment for proteins found to be more abundant in samples from patients infected with *E. faecalis*, we observed a significant enrichment in proteins associated with cholesterol metabolism. These included processes such as reverse cholesterol transport, cholesterol efflux, chylomicron, and very low-density lipoprotein particle dynamics (Fig. 4D).

In our analysis to distinguish between *E. faecalis* and *E. faecium* infections, the top proteomic biomarkers identified were immunoglobulin kappa variable 2–30 and retinol binding protein 4 (RBP4), which produced ROC AUCs of 0.76 and 0.79, respectively (Fig. 4E). Additionally, the top metabolomic features identified were retinol and PC(16:1/0:0), which had ROC AUCs of 0.77 and 0.82, respectively. The machine learning model showed

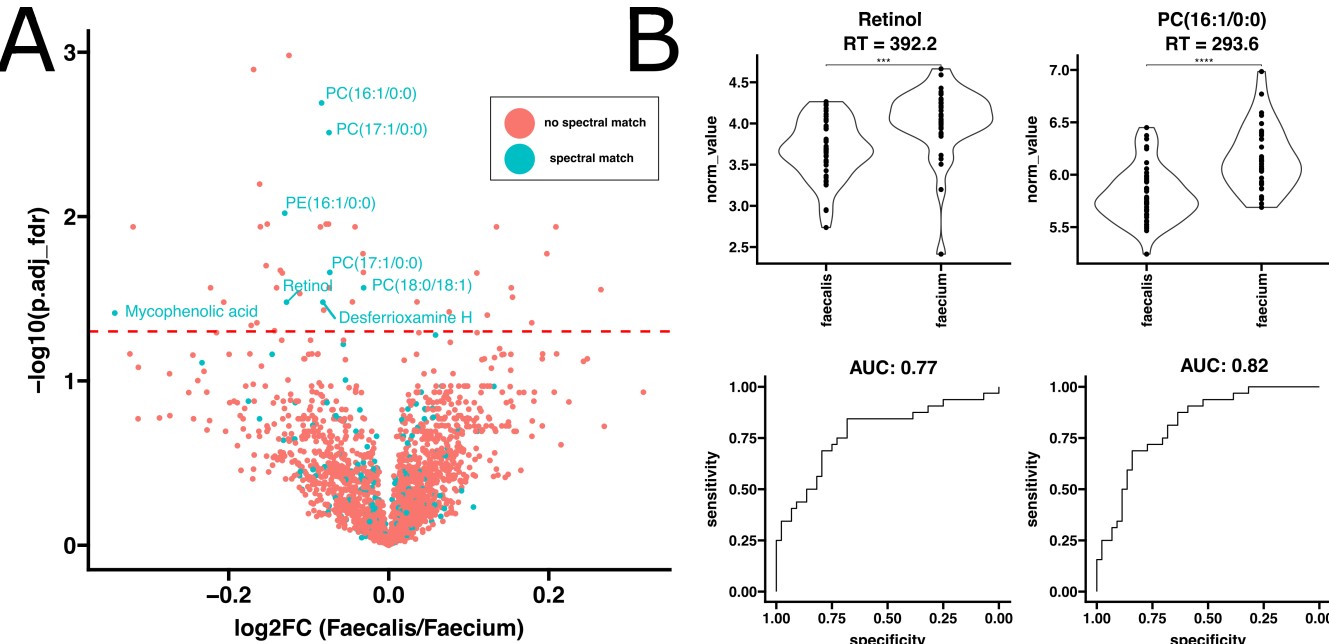

FIG 5 Untargeted metabolomics reveals differences between *Enterococcus faecalis* and *Enterococcus faecium* bacteremia. (A) Volcano plot comparing log2 fold change and FDR adjusted *P*-values resulting from *t*-tests of normalized metabolite abundances comparing *E. faecalis* to *E. faecium*. (B) Evaluation of top 2 performing putatively identified metabolite spectra matches as ranked using ensemble feature selection for distinguishing enterococcal bacteremia patients from healthy. Violin plot statistics indicate results of *t*-tests.

similar performance when evaluated on an unseen test set, achieving an ROC AUC of 0.71 (Fig. S11). There was notable overlap in the features important to the model and those identified with the alternate EFS-based approach (Fig. S5 and S11). Additionally, we noted that no significant differences were observed in the inferred cytokine profiles between *E. faecalis* and *E. faecium* (Fig. S6).

In the realm of biomarker discovery, it is important to ensure that the differences observed in biomarkers are truly attributable to the variable of interest and not confounded by additional factors (48, 49). Our evaluation of clinical metadata suggested the potential presence of confounding variables within our data set. Specifically, transplant type was significantly associated with the levels of 9 out of the top 10 ranked protein biomarkers (Fig. S7). Further investigation revealed that this association could be challenging to disentangle from the type of pathogen causing the infection, as patients infected with *E. faecium* were more likely to have organ transplants in our cohort. To address this, we refined our analysis to include only patients who had not undergone a transplant. Under these conditions, we found that antibody levels remained significantly reduced in *E. faecium*-infected samples compared to *E. faecalis* (Fig. S10A). However, we observed a loss of significance for the remaining protein biomarkers APOC1, AZGP1, PCOX1, RBP4, and SERPINC1, indicating that their significance may be confounded by transplant status (Fig. S10C). Nevertheless, the direction of enrichment for APOC3 and RBP4 was preserved, narrowly missing our threshold for statistical significance with $P$-values of 0.098, and 0.064, respectively. When examining only non-transplant patients, metabolites like PE(16:1/0:0), mycophenolic acid, and PC(16:1/0:0) no longer displayed statistically significant differences between *E. faecalis* and *E. faecium,* suggesting the observed significance may also be driven by transplant type and treatment. However, PC(16:1/0:0), PC(17:1/0:0), and retinol still showed significant differences, indicating these conclusions were not confounded by transplant status (Fig. S10D).

Additionally, smoking status was significantly associated with the abundances of 5 out of the top 10 protein biomarkers (Fig. S7). When filtering for non-smoking patients, significant differences in the levels of these biomarkers between *E. faecalis* and *E. faecium* persisted, suggesting that these biomarkers were not confounded by smoking status (Fig. S10B). Interestingly, AZGP1, a top biomarker for distinguishing *E. faecalis* from *E. faecium* and a gene reported to be overexpressed in the airway upon smoking (50), was not found to be influenced by smoking status in our data set.

A machine learning model trained solely on clinical metadata effectively utilized antibiotic sensitivity profiles to predict the type of EcB with high accuracy, achieving an ROC AUC of 0.94 on the test set (Fig. S12A). However, when features not available at the time of admission were excluded, model performance decreased significantly, resulting in an ROC AUC of 0.75 (Fig. S12B).

## Prediction of clinical outcome

Next, we set out to combine the multi-omic data with the extensive medical metadata associated from our patient cohort to identify biomarkers predictive of mortality at the time of hospital presentation and admission. Our unsupervised hierarchical clustering analyses of proteomics (Fig. 1C) or metabolomics data (Fig. 1D) revealed no high-level associations of mortality with overall proteomic or metabolomic profiles. However, when conducting feature-level analysis, we identified specific proteins that were significantly associated with mortality outcomes. We found that 84 proteins were significantly enriched in patients who died, while 25 proteins were significantly enriched in patients who survived (Fig. 6A).

The enrichment analysis of proteins significantly associated with survival revealed an association with biological processes and components linked to cellular structures and signaling functions. Specifically, the significant proteins were enriched for terms associated with the Golgi apparatus, external side of the plasma membrane, calcium ion binding, and extracellular matrix (Fig. 6B). These terms suggest a role for cellular trafficking, membrane interactions, and structural integrity in influencing patient

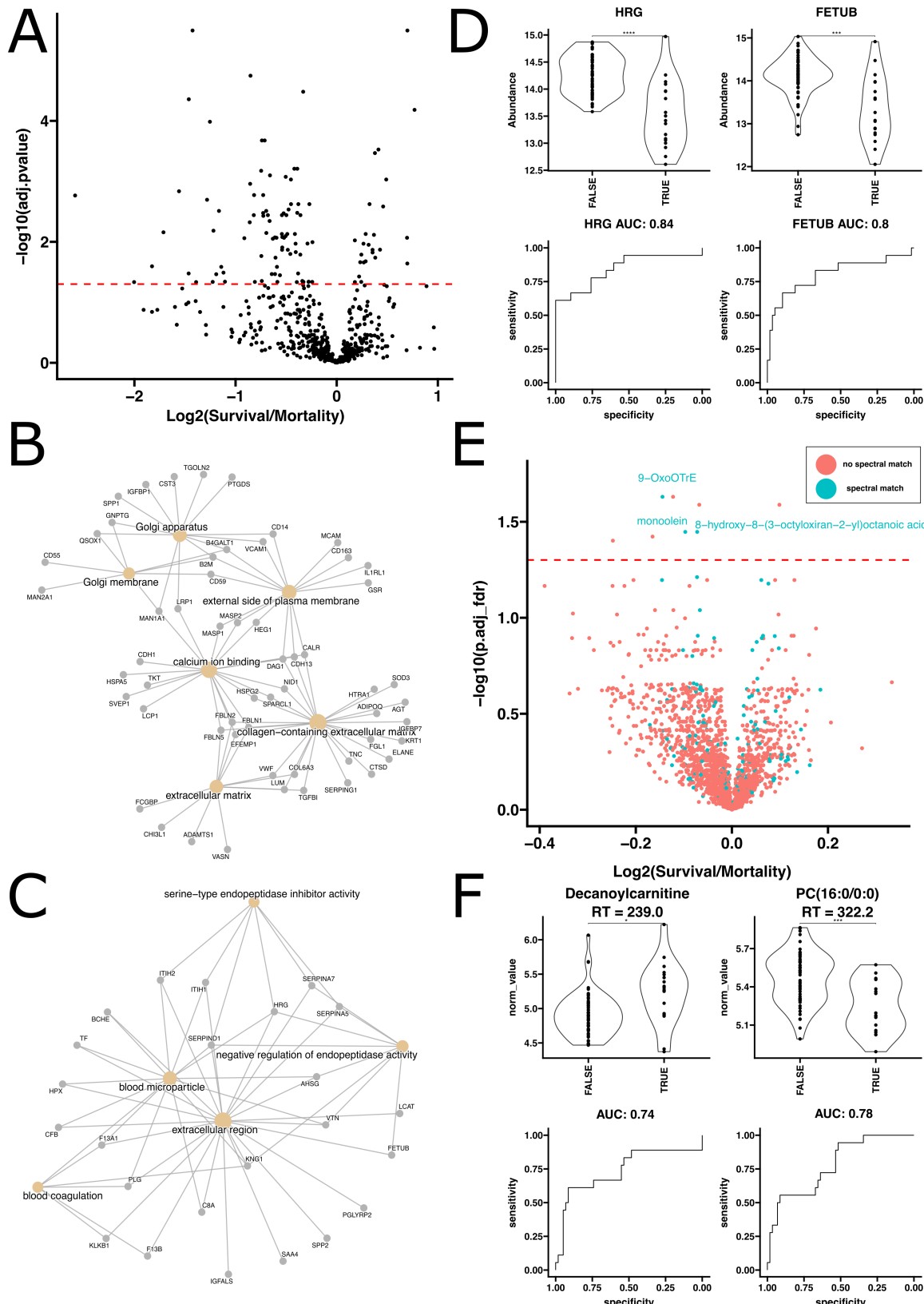

**FIG 6** Multi-omic discrimination of mortality from survival. (A) Volcano plot comparing log2 fold change and FDR adjusted *P*-values of protein abundances observed when comparing patients who suffered mortality during admission to those who survived. (B) GO term enrichment analysis of the proteins found to be significantly enriched (FDR adjusted *P*-value ≤ 0.05) in mortality relative to survival. (C) GO term enrichment analysis of the proteins found to be significantly

Fig 6 (Continued)

enriched (FDR adjusted *P*-value ≤ 0.05) in survival relative to mortality. (D) Evaluation of top 2 performing protein biomarkers as ranked using ensemble feature selection for distinguishing mortality from survival. Violin plot statistics indicate results of *t*-tests. (E) Volcano plot comparing log2 fold change and FDR adjusted *P*-values resulting from *t*-tests of normalized metabolite abundances observed when comparing patients who suffered mortality during admission to those who survived. (F) Evaluation of top 2 performing metabolite spectral matches as ranked using ensemble feature selection for distinguishing mortality from survival. Violin plot statistics indicate results of *t*-tests.

survival, potentially through mechanisms that enhance cellular resilience and communication.

Conversely, proteins significantly associated with mortality were enriched in terms related to acute response and homeostasis disruptions. Notably, these proteins were associated with blood coagulation, blood microparticle, and negative regulation of endopeptidase activity serine-type endopeptidase inhibitor activity, (Fig. 6C). This enrichment highlights the importance of coagulation processes and immune response regulation in the context of mortality, indicating that disruptions in these pathways might contribute to poorer outcomes in patients.

Histidine-rich glycoprotein (HRG) and fetuin-B (FETUB) were identified as the two best performing biomarkers for distinguishing mortality from survival in our study, showing excellent predictive accuracy when evaluated by logistic regression where they produced ROC AUCs of 0.84 and 0.80, respectively (Fig. 6D). Our complementary machine learning model showed similar performance when evaluated on an unseen test set, achieving an ROC AUC of 0.83 (Fig. S11). Once again, we noted considerable overlap in the features important to the model and those identified with the alternate EFS-based approach (Fig. S5 and S11).

In our investigation into the impact of metabolites on mortality outcomes, we found that very few metabolite features showed significant differences. Specifically, seven metabolites were significantly increased in patients who succumbed to mortality, while only two showed significant increases in patients that survived (Fig. 6E). Among these, the top two identified metabolite biomarkers, decanoylcarnitine and PC(16:0/0:0), showed moderate discriminatory accuracy, producing ROC AUCs of 0.74 and 0.78, respectively (Fig. 6F). Further assessment of our top 10 protein and metabolite biomarkers of mortality for potential confounding variables revealed no significant associations, with the exception of ICU admission status, an expected association with mortality (Fig. S7). Highlighting the degree to which ICU admission status is confounded with mortality, a machine learning model trained with clinical metadata was able to leverage ICU admission status to predict mortality with a ROC AUC of 0.91 (Fig. S13A). When ICU admission status was withheld from the model, the model lost all predictive accuracy as expected (Fig. S13B).

Lastly, we asked if there were any specific genomic features of the strains of *E. faecalis* or *E. faecium* that were associated with mortality. Using nanopore sequencing and *de novo* assembly, we analyzed the gDNA from *E. faecium* and *E. faecalis* isolates collected from patients in our study. However, hierarchical clustering based on the presence or absence of gene content across the clinical strains revealed no significant correlations with mortality outcomes (Fig. S9).

## DISCUSSION

The progress achieved in medical and surgical interventions has brought about a demographic shift toward an aging population, which often presents with increased comorbidities and compromised immune systems. Simultaneously, there is a rising prevalence of drug-resistant pathogens, necessitating the frequent use of broad-spectrum antibiotics. This combination has created a "perfect storm," paving the way for the emergence of less virulent but intrinsically antibiotic-resistant commensals, such as *Enterococcus* spp., to become significant pathogens in invasive diseases such as bacteremia. Despite the increasing relevance of enterococci as pathogens, there has

been a lack of comprehensive, unbiased research describing the host systemic response to enterococcal bacteremia using a multi-omics approach.

Beyond advancing our understanding, a comprehensive evaluation of the systemic response to EcB also serves a practical purpose. Patients with EcB often do not present with the classical signs and symptoms of infection such as fever, elevated white blood cell count, or localized inflammation and pain. Instead, their symptoms can be vague and non-specific, including generalized weakness, malaise, and weight loss, which complicates diagnosis and assessment of their condition. Currently, molecular tests that analyze the host systemic response are not commonly used in clinical settings to diagnose bacteremia or inform therapeutic decisions, However, the data from our multi-omic analysis of the systemic response to EcB provide valuable insights that could potentially be leveraged to help predict the presence of enterococcal bacteremia, distinguish whether the infection is caused by *E. faecalis* or *E. faecium*, and assess if a patient is responding well to the current treatment strategy.

In principle, molecular diagnostics that leverage differences in systemic host response have several desirable qualities. They utilize plasma, a clinically accessible and easy-to-obtain biospecimen, and can detect disease-relevant host proteins without any need for signal amplification, which is necessary in blood culture-based testing. As a result, these tests offer the potential to provide informative results within hours rather than days. If true differences in host systemic response exist, they represent attractive targets to exploit for developing novel diagnostics.

Considering these concepts, our work employs high-resolution plasma profiling using TMT proteomics and metabolomics to begin dissecting the host responses of patients afflicted with EcB, focusing on the two most common species: *E. faecalis* and *E. faecium*. This approach is driven by three overarching goals: (i) to characterize the host response EcB relative to homeostasis (i.e., healthy); (ii) to determine the systemic differences between *E. faecalis* and *E. faecium* bacteremia; (iii) to define the systemic response associated with increased mortality, offering a starting point for the future development of molecular methods that could be used to stratify patients based on predicted outcomes and subsequently intervene to improve.

We observed that the systemic response to EcB is dramatically different from homeostasis, both globally and on the scale of individual protein and metabolite features. We identified several individual protein and metabolites based on spectral matches (annotation at level 2 or 3 according to 2007 metabolomics standards initiative guidelines [51]) that were able to predict the presence of EcB relative to healthy volunteers with near-perfect discriminatory power. This indicates the potential for molecular diagnostics to predict EcB, but for these diagnostics to be more clinically useful than general molecular markers of inflammation such as CRP or SAA1, they must have some specificity to EcB rather than solely being indicators of inflammation. When we leveraged a complementary data set comparing *S. aureus* bacteremia to healthy volunteers to examine this possibility, we found several significant proteins and biological processes that differed across these two types of bacteremia, in addition to many expected conserved responses. Notably, proteins associated with the platelet alpha granule lumen were decreased in *S. aureus* bacteremia. Platelet alpha granules contain proteins with direct microbicidal properties as well as chemokine functions. Two of the proteins associated with platelet alpha granules, PPBP (CXCL7) and PF4 (CXCL4), are potent chemokines that attract neutrophils. This, combined with the observation that neutrophil-associated proteins are significantly enriched in EcB but not in *S. aureus* bacteremia, suggests differences in the role of platelet/neutrophils between EcB and *S. aureus* bacteremia and/or the differential effect of these organisms on platelet and neutrophil function (52, 53)

Although both the *S. aureus* and EcB studies were conducted in our lab using similar high-resolution TMT proteomic workflows, we cannot directly compare the two data sets due to limitations inherent in TMT-based proteomics. TMT normalization requires a "bridge" channel that consists of a pooled aliquot from all samples in the study. Since our

studies were conducted independently at different times, there was no shared pooled sample. Nevertheless, the observed differences in relation to healthy patients suggest distinct features of the host response that could help differentiate between these types of bacteremia. A larger study specifically designed to compare these and other types of bacteremia is warranted to confirm this through direct comparisons.

When we applied previously established methods of proteomics-based cytokine inference, we inferred there to be a significant increase in TNF-alpha in EcB. Notably, IL-6 showed a similar increase in EcB, but narrowly missed our threshold for statistical significance. TNF-alpha and IL-6 are two major inflammatory cytokines that are elevated in patients with bacteremia and sepsis (54), and their production by the innate immune system in response to bacterial infection is likely a major driver of the general inflammatory responses observed. As such, these cytokines are unlikely to be useful for specifically predicting EcB, instead functioning as markers of general inflammation.

As expected, several proteins involved in the acute phase response or inflammatory processes were observed to be enriched during systemic infections with both analyzed enterococcal species and *S. aureus*. Most of the features we identified as being most effective at distinguishing healthy from infected individuals have been previously reported to be biomarkers of other inflammatory processes; these include gelsolin (55, 56), leucine-rich alpha-2-glycoprotein 1 (LRG1) (57), and lipopolysaccharide binding protein (LBP) (58). Furthermore, proteins involved in cholesterol metabolism were noted to be reduced upon infection in all bacteremia types. This observation aligns with cholesterol's involvement in a myriad of biological processes, including immunity, cellular membrane functions, signaling, pathway regulation, and as a precursor for the synthesis of steroid hormones, bile acids, vitamin D, and oxysterols (59). Both low-density lipoprotein (LDL) and high-density lipoprotein (HDL) cholesterol levels are reported to be reduced in cases of sepsis, regardless of the causative organism (60), and here, we show that hypocholesterolemia is also a major feature of EcB.

Our observations that primary and secondary bile acids were significantly enriched upon infection across all bacteremia types clearly indicate cholestasis, where inflammation caused by proinflammatory cytokines results in impaired bile acid flow and increased bile acid concentrations in serum/plasma (61). This process is commonly observed during systemic inflammation, which also has the capacity to activate and amplify coagulation. Consequently, changes in proteins associated with coagulation have been reported as a general response to sepsis, regardless of the causative organism (62). We observed that several of our top identified features for distinguishing healthy from infected individuals are known to be involved in the degradation of fibrin clots or platelet aggregation. These include alpha 1-antichymotrypsin (SERPINA3) (63, 64) and enolase 1 (ENO1) (65), which were increased in infections, as well as protein C inhibitor (SERPINA5) (66, 67), CLEC3B (tetranectin) (68), and 13-OXO-ODE (69), which were decreased.

Most of the top metabolite features associated with the presence of EcB, as reported in this study, represent novel associations with bacteremia, suggesting the potential to be specific markers for EcB. Two of the top features associated with EcB, 9-oxootre and cyclo(l-phe-d-pro), have noted antimicrobial activities. 9-Oxootre, an oxylipin produced by the LOX, COX, and CYP450 pathways (70), has been shown to possess antimicrobial activity against various bacterial and fungal species (71), along with anti-inflammatory properties. Cyclo(l-phe-d-pro) is a diketopiperazine, the smallest cyclic peptides known, isolated from Gram-positive bacteria, fungi, and higher organisms (72). It has demonstrated strong antibiotic activity against *Vibrio anguillarum* (73) and likely possesses antimicrobial activity in other contexts as well. The observed reduction in these metabolites during infection could indicate the consumption of these antimicrobial metabolites produced by the host in combating the infection.

Another top biomarker, threonylcarbamoyladenosine, a nucleoside modification found in all kingdoms of life, has been noted to restrict translation initiation to the AUG start codon and suppress frameshifting at tandem ANN codons (74). This modification

has also been proposed as a strong candidate biomarker for COVID-19 infection and severity (75). To determine the specificity of these metabolite biomarkers for EcB, it would be essential to conduct a comprehensive study directly comparing plasma from multiple types of bacteremia patient populations.

In our metabolomics analysis, we note that three of the most effective metabolites at differentiating healthy from infected individuals have roles as industrial plasticizers: phthalic anhydride, 7-bis(2-ethylhexyl) phthalate, and 5-tris(2-butoxyethyl) phosphate (76, 77). These differences were likely due to logistical constraints that resulted in subtle differences in the plastics used for plasma collection between sources. This highlights the importance of seemingly innocuous components of experimental design in highly sensitive, untargeted multi-omic approaches, as has been reported elsewhere (78). These findings suggest that some metabolite features distinguishing healthy from infected samples may be confounded by differences in sample collection. Therefore, caution is advised when interpreting metabolites that show differences between healthy and infected samples in this data set. Metabolites identified as industrial chemicals, which are not typically expected to be found in blood, are likely unwanted artifacts.

Differences between *E. faecalis*-infected samples from *E. faecium*-infected samples proved to be less stark than those observed when comparing healthy to infected individuals. However, we identified several features that could distinguish the two infection types with moderate performance, correctly predicting the outcome in around ~80% of cases in our data set. Distinguishing *E. faecalis* from *E. faecium* based on host response has the potential to be a clinically important diagnostic tool, as it could inform the optimal choice of antimicrobial therapy more quickly than the current state-of-the-art methods, potentially improving patient outcomes as a result. However, our data suggest that the ability to distinguish these highly related infections based on systemic host response, as observed in plasma, is limited to a best-case scenario of around 80% accuracy. This degree of predictive power may not be sufficient for clinical use, where clinicians would certainly opt for slower, but more accurate, current methods.

Even so, our approach uncovered interesting differences between *E. faecium* and *E. faecalis* bacteremia. Most strikingly, consistent reductions in immunoglobulins were observed in *E. faecium* relative to both *E. faecalis* and healthy samples. Statistical assessment of the extensive clinical metadata collected indicated that this association was unlikely to be driven by any confounding differences, for example, organ transplant status, offering strong evidence that immunoglobulins are truly reduced in *E. faecium* bacteremia compared to *E. faecalis*. This raises the question of whether this observation is due to the manipulation of host processes by the pathogen after infection, or whether they predate the infection and instead influence susceptibility. The potential for this reduction in immunoglobulins to be a direct consequence of *E. faecium* infection exists, as there are many reports of important human pathogens reducing immunoglobulins through various mechanisms, including the direct degradation of antibodies, as demonstrated to be important virulence strategies in several bacterial species (79, 80). Alternatively, the reduction we observe in *E. faecium* bacteremia may be an indication that patients with lower titers of immunoglobulins are particularly susceptible to *E. faecium*, perhaps indicating that antibodies are more important for preventing *E. faecium* bacteremia than they are for *E. faecalis*. *E. faecium* is highly prevalent in patients with immunosuppressive diseases, notably solid organ transplant patients, supporting this notion (81). More research is necessary to conclusively determine which of these two possibilities explains the difference in immunoglobulins reported between the two cohorts.

Both the metabolomics and proteomics data pointed to differences in retinol (vitamin A) abundance and transport between the two types of EcB, with both retinol and RBP4 being increased in *E. faecium* bacteremia relative to *E. faecalis*. These were among the top proteomic and metabolomic features capable of discriminating the infections. While RBP4 levels were potentially confounded by transplant status in our data set, narrowly missing our threshold for significance when considering only patients

without a transplant, retinol levels remained significantly different. This suggests that the differences in these retinol-associated features were most likely due to *E. faecalis* and *E. faecium*. Retinol levels decline during the acute-phase response to infection as a consequence of reduced RBP transcription in the liver (82) and increased urinary loss, suggesting that the acute-phase response to *E. faecium* bacteremia may be muted relative to what is observed in *E. faecalis*-driven bacteremia. Retinol is important for the function of various aspects of the innate and adaptive immune system, and the differences observed in this study may impact immune system function (83). Perhaps as a consequence of a reduced acute-phase response, we also noted significant decreases in proteins associated with cholesterol metabolism in *E. faecalis* compared to *E. faecium* bacteremia and healthy volunteers. While the suppression of serum lipoproteins in response to infection has been reported previously, differences have never been reported between two types of closely related bacteremia. It is possible that these differences function as an indicator of the underlying disease severity, as lipoprotein levels in serum have been shown to correlate with infection severity (84), and *E. faecium* tends to be less virulent than *E. faecalis* (85). If this were indeed the case, we would expect to see lipoproteins also associated with mortality, a finding not observed in our study. Thus, the underlying reasons for these interesting differences in proteins involved in cholesterol metabolism remain unclear.

The evaluation of the systemic responses comparing mortality to survival uncovered several important differences among these patients. We found that features derived from proteomics data were more effective at predicting mortality status than those derived from metabolomics data. The biological processes observed to decrease in patients who succumbed to mortality, compared to those who survived, primarily consisted of the same ones found to be significantly different when comparing healthy individuals to those infected. This suggests that the degree of alteration relative to baseline for proteins involved in blood coagulation, blood microparticles (lipoproteins), and negative regulation of endopeptidase (SERPINS) activity correlates with the severity of EcB. Conversely, the biological processes that were found to be significantly increased in mortality were vaguer, encompassing calcium ion binding, the extracellular matrix, the external side of the plasma membrane, and the Golgi apparatus. The potential biological relevance of these increases is challenging to discern, indicating that while they are associated with mortality, their specific roles in the pathophysiology of the disease remain unclear and require further investigation.

A number of features from the proteomics data were able to predict mortality, with classification accuracies greater than 80%. Notably, decreases in two cysteine protease inhibitors belonging to the type-3 cystatins class of structurally and functionally related proteins (86), HRG and FETUB, were the best predictors of mortality. These proteins have roles in blood coagulation, and HRG has been shown to possess antibacterial properties (87). They have also been identified as biomarkers of mortality in various studies, with evidence indicating that lower levels are associated with mortality in conditions such as COVID-19 (88), *S. aureus* bacteremia (18), as even in mouse models of sepsis, where administration of exogenous HRG was able to improve outcomes (89). This evidence suggests that a similar therapeutic approach involving HRG may be beneficial in the context of EcB.

The degree of accuracy displayed by these potential biomarkers could be clinically relevant, especially since there is currently no existing quantitative method to stratify patients specifically in the setting of EcB. Diagnostic tests that achieve ROCs ranging from 0.8 to 0.9 are considered to have excellent discriminatory power (90), indicating that the accuracy levels we observed could be highly useful in clinic settings. Further research is necessary to determine whether the predictive accuracy observed in this study can be replicated in an independent cohort. Additionally, it would be important to investigate whether predicting patient outcomes can be effectively paired with targeted interventions.

In conclusion, our study has significantly advanced a powerful multi-omics framework, as outlined in previous research (18) to produce the most granular profile of the systemic response to EcB to date. We have reported significant differences between the multi-omic plasma profiles of EcB relative to healthy volunteers, differences driven by infections with *E. faecalis* versus *E. faecium*, and distinctions between outcomes related to mortality and survival. These data are easily explorable at https://gonzalezlab.shinyapps.io/EcB_multiomics/. The culmination of these efforts represents a significant stride toward characterizing the systemic response to EcB and its relationship to other bacteremia types, while enhancing future abilities to risk-stratify patients for improved antimicrobial and/or immunotherapy approaches.

## ACKNOWLEDGMENTS

C.B. and L.A.-R. were supported in part by the UCSD Graduate Training Program in Cellular and Molecular Pharmacology through an institutional training grant from the National Institute of General Medical Sciences, T32 GM007752. C.S. was supported in part by the NIH/NIAMS T32 AR064194 Rheumatic Diseases Training Grant. D.J.G. is supported by R01AI148417. This study was also supported by the UCSD Collaborative Center of Multiplexed Proteomics.

## AUTHOR AFFILIATIONS

[1]Biomedical Sciences Graduate Program, UC San Diego, La Jolla, San Diego, California, USA

[2]Department of Pharmacology, University of California San Diego, La Jolla, San Diego, California, USA

[3]Skaggs School of Pharmacy and Pharmaceutical Sciences, University of California San Diego, La Jolla, San Diego, California, USA

[4]Department of Pharmacy, UW Health, Madison, Wisconsin, USA

[5]School of Pharmacy, University of Wisconsin-Madison, Madison, Wisconsin, USA

[6]Department of Pediatrics, UC San Diego, La Jolla, San Diego, California, USA

[7]Center for Microbiome Innovation, University of California at San Diego, La Jolla, San Diego, California, USA

[8]Sharp Rees Stealy Medical Group, San Diego, California, USA

## AUTHOR ORCIDs

Charlie Bayne  http://orcid.org/0000-0003-1870-5298
Leigh-Ana Rossitto  http://orcid.org/0000-0002-7744-3630
Carlos Gonzalez  http://orcid.org/0000-0002-4673-4048
Haoqi Nina Zhao  http://orcid.org/0000-0003-3908-630X
Victor Nizet  http://orcid.org/0000-0003-3847-0422
George Sakoulas  https://orcid.org/0000-0002-6063-6217
David J. Gonzalez  http://orcid.org/0000-0003-1423-5970
Warren Rose  http://orcid.org/0000-0001-5012-5993

## FUNDING

| Funder | Grant(s) | Author(s) |
| --- | --- | --- |
| HHS | NIH | National Institute of General Medical Sciences (NIGMS) | GM007752 | Charlie Bayne |
|  |  | Leigh-Ana Rossitto |
| HHS | NIH | National Institute of Arthritis and Musculoskeletal and Skin Diseases (NIAMS) | AR064194 | Concepcion Sanchez |
| HHS | NIH | National Institute of Allergy and Infectious Diseases (NIAID) | R01AI148417 | David J. Gonzalez |

## AUTHOR CONTRIBUTIONS

Charlie Bayne, Conceptualization, Data curation, Formal analysis, Investigation, Methodology, Validation, Visualization, Writing – original draft, Writing – review and editing | Dominic McGrosso, Conceptualization, Formal analysis, Investigation, Methodology, Visualization, Writing – original draft, Writing – review and editing | Concepcion Sanchez, Formal analysis, Methodology, Visualization | Leigh-Ana Rossitto, Formal analysis, Methodology | Maxwell Patterson, Formal analysis, Visualization, Writing – review and editing | Carlos Gonzalez, Funding acquisition, Methodology, Supervision, Writing – review and editing | Courtney Baus, Data curation, Methodology | Cecilia Volk, Investigation, Methodology | Haoqi Nina Zhao, Investigation, Methodology | Pieter Dorrestein, Data curation, Funding acquisition, Methodology, Resources | Victor Nizet, Conceptualization, Funding acquisition, Methodology, Writing – review and editing | George Sakoulas, Conceptualization, Methodology, Project administration, Resources, Writing – review and editing | David J. Gonzalez, Conceptualization, Investigation, Methodology, Project administration, Resources, Supervision, Writing – review and editing | Warren Rose, Conceptualization, Funding acquisition, Methodology, Project administration, Resources, Supervision, Writing – review and editing

## DATA AVAILABILITY

The data reported in this manuscript are available on MassIVE under the following identifiers: proteomics, MSV000096728; metabolomics, MSV000096729. All analysis of data downstream of Fragpipe (proteomics) or MzMine3 (metabolomics) processing can be found in the GitHub repo at https://github.com/baynec2/EcB_multiomics. An interactive data analysis app to explore the data contained in this paper can be found hosted at https://gonzalezlab.shinyapps.io/EcB_multiomics/. The source code to build the shiny web application can be found in the 04_shiny_app directory within the GitHub repo. Any other data are available upon request.

## ETHICS APPROVAL

All human samples were approved by UW Madison/UW Health under protocol number 2018-0098.

## ADDITIONAL FILES

The following material is available online.

### Supplemental Material

**Figure S2 (mSystems01471-24-S0001.pdf).** Functional networks of significantly different features.
**Supplemental Material (mSystems01471-24-S0002.docx).** Supplemental figures and captions for supplemental tables.
**Table S1 (mSystems01471-24-S0003.xlsx).** Protein level comparisons between groups.
**Table S2 (mSystems01471-24-S0004.xlsx).** Metabolite comparisons between groups.

### Open Peer Review

**PEER REVIEW HISTORY (review-history.pdf).** An accounting of the reviewer comments and feedback.

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
