## [Reviewer comments · mSystems]

Multi-omic Signatures of Host Response Associated with Presence, Type, and Outcome of Enterococcal Bacteremia

Charlie Bayne, Dominic McGrosso, Concepcion Sanchez, Leigh-Ana Rossitto, Maxwell Patterson, Carlos Gonzalez, Courtney Blaus, Cecilia Volk, Haoqi Nina Zaho, Pieter Dorrestein, Victor Nizet, George Sakoulas, David Gonzalez, and Warren Rose

Corresponding Author(s): Warren Rose, University of Wisconsin-Madison School of Pharmacy

Review Timeline:

Submission Date:

November 2, 2024

Accepted:

December 6, 2024

Editor: Aleksandra Nita-Lazar

Reviewer(s): Disclosure of reviewer identity is with reference to reviewer comments included in decision letter(s). The following individuals involved in review of your submission have agreed to reveal their identity: Malgorzata Barbara Łobocka (Reviewer #2)

Transaction Report:

DOI: <https://doi.org/10.1128/msystems.01471-24>

Re: mSystems01471-24 (Multi-omic Signatures of Host Response Associated with Presence, Type, and Outcome of Enterococcal Bacteremia)

Dear Dr. Warren E. Rose:

Your manuscript has been accepted, and I am forwarding it to the ASM production staff for publication. Your paper will first be checked to make sure all elements meet the technical requirements. ASM staff will contact you if anything needs to be revised before copyediting and production can begin. Otherwise, you will be notified when your proofs are ready to be viewed.

Sincerely,
Aleksandra Nita-Lazar
Editor
mSystems

Reviewer #1 (Comments for the Author):

the revision addressed our concerns.

Reviewer #2 (Comments for the Author):

The revised version of the manuscript has been corrected according to the reviewers recommendations. The authors addressed all my comments. I have only two minor comments concerning cosmetic corrections.

L. 220: Replace "minute s" with "minutes"

L. 664 "ad"?